# The Novel Achievements in Oncological Metabolic Radio-Therapy: Isotope Technologies, Targeted Theranostics, Translational Oncology Research

**DOI:** 10.3390/medsci13030107

**Published:** 2025-08-01

**Authors:** Elena V. Uspenskaya, Ainaz Safdari, Denis V. Antonov, Iuliia A. Valko, Ilaha V. Kazimova, Aleksey A. Timofeev, Roman A. Zubarev

**Affiliations:** 1Department of Pharmaceutical and Toxicological Chemistry, Medical Institute, Peoples’ Friendship University of Russia Named After Patrice Lumumba (RUDN University), 6 Miklukho-Maklaya St., Moscow 117198, Russia; ainazsafdari98@gmail.com (A.S.); 1142240239@pfur.ru (D.V.A.); kazimova.ilaha96@gmail.com (I.V.K.); zubarev_ra@pfur.ru (R.A.Z.); 2Department of Clinical Immunology, Allergology and Adaptology, Faculty of Continuous Medical Education, Peoples’ Friendship University of Russia Named After Patrice Lumumba (RUDN University), 21/3 Miklukho-Maklaya St., Moscow 117198, Russia; valko_yulya@mail.ru; 3Scientific and Educational Resource Centre “Innovative Technologies of Immunophenotyping, Digital Spatial Profiling and Ultrastructural Analysis”, Peoples’ Friendship University of Russia Named After Patrice Lumumba (RUDN University), 6 Miklukho-Maklaya Street, Moscow 117198, Russia; alexpismo77@mail.ru; 4Division of Chemistry I, Department of Medical Biochemistry and Biophysics, Karolinska Institutet, 17177 Stockholm, Sweden; 5Chemical Proteomics Unit, Science for Life Laboratory (SciLifeLab), 17165 Stockholm, Sweden

**Keywords:** cancer epidemiology, targeted nuclides, radiotherapy, theranostics, peptide receptor radionuclide therapy, mABs-drug, target complexes, translational oncology

## Abstract

Background/Objectives. This manuscript presents an overview of advances in oncological radiotherapy as an effective treatment method for cancerous tumors, focusing on mechanisms of action within metabolite–antimetabolite systems. The urgency of this topic is underscored by the fact that cancer remains one of the leading causes of death worldwide: as of 2022, approximately 20 million new cases were diagnosed globally, accounting for about 0.25% of the total population. Given prognostic models predicting a steady increase in cancer incidence to 35 million cases by 2050, there is an urgent need for the latest developments in physics, chemistry, molecular biology, pharmacy, and strict adherence to oncological vigilance. The purpose of this work is to demonstrate the relationship between the nature and mechanisms of past diagnostic and therapeutic oncology approaches, their current improvements, and future prospects. Particular emphasis is placed on isotope technologies in the production of therapeutic nuclides, focusing on the mechanisms of formation of simple and complex theranostic compounds and their classification according to target specificity. Methods. The methodology involved searching, selecting, and analyzing information from PubMed, Scopus, and Web of Science databases, as well as from available official online sources over the past 20 years. The search was structured around the structure–mechanism–effect relationship of active pharmaceutical ingredients (APIs). The manuscript, including graphic materials, was prepared using a narrative synthesis method. Results. The results present a sequential analysis of materials related to isotope technology, particularly nucleus stability and instability. An explanation of theranostic principles enabled a detailed description of the action mechanisms of radiopharmaceuticals on various receptors within the metabolite–antimetabolite system using specific drug models. Attention is also given to radioactive nanotheranostics, exemplified by the mechanisms of action of radioactive nanoparticles such as Tc-99m, AuNPs, wwAgNPs, FeNPs, and others. Conclusions. Radiotheranostics, which combines the diagnostic properties of unstable nuclei with therapeutic effects, serves as an effective adjunctive and/or independent method for treating cancer patients. Despite the emergence of resistance to both chemotherapy and radiotherapy, existing nuclide resources provide protection against subsequent tumor metastasis. However, given the unfavorable cancer incidence prognosis over the next 25 years, the development of “preventive” drugs is recommended. Progress in this area will be facilitated by modern medical knowledge and a deeper understanding of ligand–receptor interactions to trigger apoptosis in rapidly proliferating cells.

## 1. Introduction

According to data from the International Agency for Research on Cancer (IARC https://gco.iarc.fr/today/en, accessed on 11 January 2025) as of 2022, approximately 20 million new cancer cases and 9.7 million cancer-related deaths were registered worldwide, including non-melanoma skin cancers (NMSCs) [1]. Data analysis shows [2] that cancer is diagnosed in about one in five individuals of both sexes—men and women—during their lifetime, while approximately one in nine men and one in twelve women die from it. Malignant neoplasms of the lung and breast (in women) are the most common types of cancer globally (12.4% and 11.6% of all new cases, respectively), colorectal cancer accounts for 9.6% in both sexes, prostate cancer for 7.3% in men, and stomach cancer for 4.9% in both sexes. Data visualization illustrating the incidence, mortality, and prevalence of 36 specific cancer types across 185 countries by sex and age group is presented in Figure 1 within the GLOBOCAN project (https://www.uicc.org/, accessed on 11 January 2025) [3].

Prognostic models predict an increase in cancer cases to 35 million by 2050, particularly affecting countries with low and medium Human Development Index (HDI), emphasizing the need for access to cost-effective cancer treatment services and the importance of developing innovative solutions in medicine and rehabilitation [4].

The evolution of cancer therapy is not confined to the 20th century, despite notable breakthroughs such as the discovery of the radioactive elements radium and polonium by Marie and Pierre Curie, marking the beginning of radiotherapy [5]; combination immunotherapy for cancer by Paul Ehrlich’s, who coined the term “chemotherapy” [6]; and the discovery of new properties of cytotoxic agents exemplified by chemical warfare mustard agents—sulfur mustard (SM, 2,2′-dichloroethyl sulfide)—and the development of one of the first alkylating chemotherapeutic drugs, nitrogen mustard (NM, mechlorethamine), by Louis Goodman and Alfred Gilman [7,8,9]. Radiotherapy has been used since 1950, following the introduction of cobalt teletherapy, which enabled precise radiation delivery to tumors while minimizing damage to surrounding healthy tissue and facilitating combination with other treatment modalities [10].

A significant improvement in patients’ quality of life and survival was driven by a dramatic shift in the cancer treatment landscape in the 21st century: theranostic methods involving conjugation of diagnostic radiopharmaceuticals with therapeutic agents; the implementation of gene therapy and nanomedicine combined with targeted therapy and efficient biodistribution of chemotherapeutic agents; delivery methods for small interfering RNA (siRNA); development of a monoclonal antibody («mAB_s_-drug»), thermal ablation and magnetic hyperthermia techniques; use of natural antioxidants in cancer therapy; as well as innovative approaches such as radiomics (quantitative assessment of tumor characteristics) and pathomics (high-resolution tissue image analysis) (Figure 2) [11,12,13,14,15,16,17,18,19].

Special attention from researchers and clinical practitioners is given to radiopharmacy, targeted radionuclide therapy, and the development and implementation of rapid diagnostic systems and compact measuring devices for determining isotope content in medical applications. This focus is driven by the advent of personalized medicine (https://www.genome.gov/genetics-glossary/Personalized-Medicine, accessed on 18 January 2025), whose core concept is the use of an individual’s genetic profile to inform decisions regarding disease prevention, diagnosis, and treatment [20,21].

## 2. Methods

The methodology for this review article involved developing our own search algorithm, which was based on the selection and analysis of information from the PubMed, Scopus, and Web of Science databases over the past 20 years. This approach was guided by the concept of the existing “structure–mechanism–effect” relationship of active pharmaceutical ingredients (API), encompassing various forms such as atoms, solid-phase (nano-) objects (d~1–100 nm), molecules, and complex particles.

In addition to review articles, practical research, and clinical cases, data from accessible official online sources were also considered, including the International Agency for Research on Cancer, the GLOBOCAN project, the National Human Genome Research Institute, the International Atomic Energy Agency, the Nuclear Research and Consultancy Group, Melanoma Unit, and NanoTherm.

Among the literary sources reviewed, articles from the last 5 years constituted approximately 60%, those from the last 20 years accounted for about 20%, and approximately 20% were older sources deemed valuable for the history of medicine.

The selection process involved screening abstracts and, more extensively, the full texts of articles. The final presentation of the material was conducted using a narrative synthesis approach, which included a systematic review and generalization of results from multiple studies, accompanied by our own conclusions. This approach was also applied to the presentation of graphic material.

## 3. Results

### 3.1. Isotope Technology—Nucleus Stability/Instability

Radiopharmacy utilizes two classes of isotopes for therapeutic purposes: stable isotopes, which do not undergo radioactive decay over time, and unstable isotopes (radioisotopes)—nuclei that undergo spontaneous radioactive decay, emitting ionizing radiation in order to transform into a stable form (https://www.iaea.org/ru/temy/radioizotopy, accessed on 25 January 2025). Nuclear stability is achieved through alpha, electron, or positron emission (in the form of gamma rays) [22,23]. There are three types of particulate radiation relevant to targeted radiopharmacy: alpha particles, beta particles, and Auger electrons, which can irradiate tissue volumes at multicellular, cellular, and subcellular scales (Table 1).

The rupture of the DNA double helix (DSB) is considered the main and most destructive mechanism of damage to tumor cells. The foremost cause of DSB is replication across a nick, giving rise to chromatid breaks during S phase [27].

Stable isotopes are primarily used as tracers in pharmacokinetic studies, for example, to investigate human metabolism kinetics in vivo. The high sensitivity and specificity of tracers allow them to be tracked through complex processes of substance redistribution and transformation, including within living organisms. In human metabolism research, the most commonly used stable isotopes are those of hydrogen, oxygen, and nitrogen (^13^C, ^15^N, ^2^H и ^18^O), which can be incorporated into molecules and used as metabolic indicators [28]. It is believed that nuclear stability is maintained by boson pairing between protons and neutrons [29]. Disruption of boson pairing leads to instability. The so-called “proton stability boundary” demonstrates a narrow linear relationship for nuclei considered stable [30]. Such nuclei are characterized by the ratio of the number of neutrons «n» to protons «p» (Equation (1)):n/p = 0.98 + 0.015·A^2/3^,(1)
where A = n + p is the mass number.

The most stable nuclei are the so-called “magic nuclei”, where the number of protons or neutrons corresponds to one of the magic numbers: 2, 8, 20, 28, 50, 82, and 126 [31].

To date, more than 3500 unstable isotopes are known, of which around 80 occur naturally, about 200 have been artificially created, and fewer than 50 are regularly used in clinical practice. The production of radiopharmaceuticals requires a nuclear infrastructure encompassing the entire radionuclide manufacturing process (Nuclear Research and Consultancy Group (https://www.advancingnuclearmedicine.com/, accessed on 28 January 2025). Medical isotopes are produced either in nuclear reactors or in cyclotrons—cyclic accelerators of non-relativistic heavy charged particles. In reactors, nuclear fission chain reactions occur in a controlled and stable environment. Reactor fuel consists of low-enriched uranium U-235, which, upon fission, produces a neutron cloud. When neutrons collide with stable isotopes, radionuclides are generated, for example, the neutron-rich «parent» atom Mo-99. Reactor and cyclotron isotopes used in radiopharmacy are listed in Table 2.

Despite years of experience in harnessing nuclear reactions for medical applications, both production routes—reactor and cyclotron—have limitations and drawbacks [34]. These include challenges in achieving high radionuclide purity and specific activity, the time-consuming and costly separation of therapeutic nuclides from radioactive impurities, limited availability due to reliance on historical uranium, actinium sources, or nuclear waste, which represent expensive investments, as well as the need for scale-up in the radiopharmaceutical market for small-batch GMP production [35].

Despite the advantages of targeted therapies, characterized by low toxicity profiles, they often exhibit low response rates to a single active pharmaceutical ingredient (API). Moreover, the emergence of resistance to targeted therapy is a significant clinical challenge, especially in patients with advanced tumors [36]. Therefore, there is a clear need for new translational strategies and targeted approaches to cancer treatment, particularly to overcome resistance.

### 3.2. Targeted Theranostics

The term «theranostics», a combination of «therapeutic» and «diagnostic» was first introduced into nuclear medicine in 1998 at the intersection of precision and personalized medicine [37]. Theranostics combines diagnosis and treatment for continuous medical assessment of a patient’s condition using an appropriate combination of active pharmaceutical and radiopharmaceutical ingredients, as well as nuclear medical imaging methods: radiotracers, contrast agents, positron emission tomography (PET), and magnetic resonance imaging (MRI) (Figure 3).

A recent clinical study highlighted the ability of alpha-radiotherapy with high linear energy transfer (LET) (see Table 1) to overcome treatment resistance to beta-particle therapy [38]. This approach allows the selection of the sub-population of patients most likely to benefit from a targeted therapy in accordance with their “molecular profile” at a given time point, or, conversely, those patients for whom the risk of adverse effects is higher [39].

#### 3.2.1. Radiopharmaceuticals—Antimetabolites

While «theranostics» entered nuclear medicine terminology relatively recently, the concept of nuclear theranostics was introduced in 1943 by Dr. S. M. Seidlin at Montefiore Hospital in New York City, who first used iodine-131 for diagnostic imaging, target expression confirmation, and radionuclide therapy of thyroid cancer in his patient [40]. In modern nuclear medicine, the combination of radionuclide I-131 and I-131-iodine-meta-iodobenzylguanidine (MIBG, Iobenguane, Azedra^R^). Iobenguane I-131—is a radiopharmaceutical ingredient representing a structural analog of the neurotransmitter noradrenaline, containing radioactive I-131 in the meta position relative to the alkylguanidine side chain (Figure 4). Iobenguane may be used to image or eradicate neuroendocrine tissues and tumor cells.

The mechanism of action of MIBG theranostics is as follows: acting as a structural analog of the natural metabolite noradrenaline, meta-iodobenzylguanidine interacts with adrenergic receptors in adrenal, liver, heart, and spleen tissues, blocking them and thereby inhibiting signal transmission, thus exhibiting antimetabolite properties. MIBG is used for diagnosing primary and metastatic pheochromocytoma or neuroblastoma. The radiopharmaceutical was approved by the FDA on 19 September 2008.

The radioisotope I-131, as the therapeutic component in this theranostic, destroys tissues metabolizing noradrenaline, undergoing transformations (Equation (2)):(2)I53131→Xe54131+e−+ve¯

Combination of metabolic radiotherapy (MtRth) with other treatment methods is the most important direction for increasing the efficiency of cancer treatment and reducing the frequency of side effects in clinical practice. This goal can be achieved by combining MtRth with neoadjuvant (NCRT) and adjuvant chemoradiotherapy (ACRT) before surgery, for example, immunotherapy, proton beam therapy, hyperthermia, etc. [41].

#### 3.2.2. Radionuclide Therapy with Peptide Receptors

Another example of a theranostic radionuclide of interest as a suitable combination of active radiopharmaceutical ingredients (ARPI) for diagnosis and therapy of neuroendocrine tumors (NETs), adenocarcinoma variants is lutetium-177.


*^177^Lu–Dotatate radioligand therapy (RLT)*


^177^Lu–Dotatate (Lutathera^R^) («Dota»—dodecanetetraacetic acid and «tate»—shortly somatostatin receptors) is a complex of a somatostatin-like peptide hormone (SST) and the octadentate ligand DOTA with a lutetium-177 central radionuclide atom (Figure 5) [42].

The pioneer of NET radiotherapy was Professor Eric Krenning at the Erasmus Medical Centre Rotterdam, who first presented clinical results in 2017 for peptide receptor radionuclide therapy (PRRT) of gastroenteropancreatic (GEP) NETs [43]. Neuroendocrine tumors of the midgut represent the most common type of malignant gastrointestinal neuroendocrine tumors and are associated with 5-year survival rates of less than 50% among persons with metastatic disease. ^177^Lu–Dotatate, as a radioactively labeled somatostatin analog, enables targeted delivery of radiation with a high therapeutic index to tumors expressing somatostatin receptors [44]. An alternative name for therapy with a similar mechanism is «radioligand therapy of somatostatin receptor» (RTSR) which has demonstrated prolonged median progression-free survival in heterogeneous NET patient populations [45].

The most commonly used chelating agents for complex formation with radioactive metals, apart from DOTA, are NOTA (1,4,7-triazacyclononane-1,4,7-triacetic acid) and NODAGA glutaric-derivatives (Figure 6) [46,47].

Target complexes based onNOTA/NODAGA/DOTA/DODAGA also showed high affinity and selectivity for GRPR (Gastrin-Releasing Peptide Receptor), SSTR2 (Somatostatin Receptor Subtype 2) and MC1R (Melanocortin-subtype 1 receptor), receptors present on the surface of primary and secondary tumors of the prostate, mammary glands, pancreas, lungs, etc. [48].

Overall, PRRT with ^177^Lu-Dotatate is an effective treatment for patients with progressive, high-grade NETs, with an approximately 80% reduction in the risk of progression or death. Monitoring of renal and blood function during and after therapy is recommended to minimize this risk [49].


*^177^Lu—vipivotide tetraxetan RLT*


Lutetium-177 vipivotide tetraxetan (Pluvicto^R^) (“vipivo”- targeting moiety Lys-Urea-Glu, the “tide” suffix—peptide nature of this moiety and “tetraxetan” is a DOTA derived from)—is a RLT drug first approved by the FDA on 23 March 2022, for the treatment of prostate-specific membrane antigen-positive metastatic castration-resistant prostate cancer (NIH, https://www.cancer.gov/publications/dictionaries/cancer-drug/def/lutetium-lu-177-vipivotide-tetraxetan, accessed on 16 April 2025). It is generally accepted that the mechanism of action of ^177^Lu-PSMA-617 is attributed to its radioligand activity [50]. The structure comprises the main fragments: radionuclide (Lu-177)—a source of β^−^ radiation; targeting ligand (vipivotide)—a PSMA-binding peptide (Lys-Urea-Glu) that specifically targets prostate cancer cells, as PSMA is significantly overexpressed on their surface; chelator (tetraxetan)—a chemical moiety that securely binds the radionuclide to the targeting ligand; hydrophobic linker, composed of 2-naphthyl-L-Ala and cyclohexyl groups, it connects the targeting ligand to the chelator, influencing the compound’s pharmacological properties (Figure 7).

The results of ^177^Lu-vipivotide tetraxetan therapy show a pronounced biochemical (decrease in the level of total PSA) response, as well as low toxicity (often in the form of the development of grade I xerostomia) [51].


*MC1R targeting. Radioactive theranostics of melanoma*


Skin cancer is the most common form of cancer, with melanoma being the most dangerous type. According to the World Health Organization, over 130,000 new cases of melanoma are diagnosed worldwide annually (https://melanomaunit.ru/vse-o-melanome/statistika/, accessed on 2 May 2025). Melanoma develops from pigment-producing cells—melanocytes—due to their malignant progression and early haematogenous and lymphogenous metastasis (metastatic melanoma, MtMn) [52]. A distinctive feature of MtMn is the elevated expression of the endocytic receptor Melanocortin 1 (MC1R) on the surface of human melanoma cells, making it a crucial tumor marker. The MC1R receptor is a melanocortin peptide that is mediated by G protein-coupled receptors (GPCRs) with an N-linked glycosylation site on its extracellular terminus and a palmitoylation site on the intracellular C-terminus. The extracellular N-terminal tail acts as a signaling anchor and plays a vital role in ligand affinity (Figure 8) [53].

Therapy for MtMn generally aims to detect and subsequently exert cytostatic effects on secondary tumors (as a consequence of advanced melanoma). A theranostic approach is employed, based on targeted delivery of a radioactively labeled peptide (active pharmaceutical ingredient, antibody-based API) to the tumor, where the highly overexpressed MC1R is activated on the cell surface [54]. In this case, the selective peptide binding domain to the cell surface antigen is identified using a native related peptide (Figure 9).

Deposition of high LET radiation over a short path length leads to increased frequency of DSB and specific destruction of MtMn tumor cells. Quantitative radiolabeling is typically achieved using elementally matched theranostic radioisotope pairs, such as ^203^Pb/^212^Pb (diagnostic/therapeutic) (see Table 2) [55,56,57]. The radioisotope Pb-212, as the therapeutic component in this theranostic, is obtained via decay transformations (Equation (3)) [58]:(3)T228h⟶R224a⟶R220n⟶P216o⟶P212b⟶B212i1.91 year   3.66 days    55.6 s 0.14 s 10.6 h  60.6 months

#### 3.2.3. Radionuclide Therapy with Hormone Receptors


*^18^F-Fluoroestradiol*


In May 2020, after decades of research, the United States Food and Drug Administration (FDA) approved the PET tracer radiopharmaceutical 6α-fluoro-17β-estradiol (^18^F-fluoroestradiol, FES) for clinical use in patients with estrogen receptor (ER)/progesterone receptor (PR)-positive recurrent or metastatic breast cancer as a complement to biopsy [59].

^18^F-fluoroestradiol binds to hormone receptors (HR) in the nuclei of ER-expressing cells, including those in the uterus or ovaries, enabling in vivo assessment of ER/PR expression throughout the body [60]. This distinguishes ^18^F-fluoroestradiol oт ^18^F-fluorodeoxyglucose (FDG, the glucose analog), which has limited sensitivity for detecting primary breast tumors [61].

The molecular structures of estradiol and ^18^F-fluoroestradiol explain the mechanism of detecting breast cancer lesions or other estrogen receptor-positive organs (Figure 10).

It can be seen that ^18^F-fluoroestradiol is a fluorinated derivative of the estrogenic steroid at the C18 position of hydrogenated cyclopentaphenanthrendiol, which explains the affinity and binding to estrogen receptors, allowing PET imaging of the lesions. ^18^F-fluoroestradiol has been used as a research agent since the 1980s and as a clinical agent since 2016 in France and 2020 in the United States, without any major adverse events reported to date [62].

#### 3.2.4. Radiopharmaceuticals—Metabolites

Delivery of receptor-targeted APIs acts as an appealing strategy for cancer treatment and diagnosis. A notable example of a combination of a natural metabolite (folate) and an antitumour agent (vinblastine) is the vintafolide/etarfolatide couple (Figure 11).

Vinca alkaloids—a class of antitubulin agents acting at the G2-M metaphase of the cell cycle—are plant-derived alkaloids with cytostatic activity. Vinca alkaloids, derived from *Vinca rosea* or *Catharanthus roseus*, include first-generation compounds (vincristine and vinblastine), semisynthetic second-generation derivatives (vinorelbine and vindesine), and third-generation agents (vinflunine). Incorporating folic acid into the complex with the API and carrier ensures specific, targeted interaction with tumor cells, due to the expression of folate receptor beta (FR-α,-β), including on cancer cells [63]. Consequently, FRs may represent targets for specific delivery of therapeutic agents to activated. Moreover, studies indicate a positive correlation between FR-β expression on mesenchymal stem cells (MSC), cancer stage, and lymph node metastases [64].

Vintafolide is a folate-targeted (FR-α) chemotherapeutic conjugate (folate-vitamin B9 + vinca alkaloid) in clinical stage development as a treatment for folate receptor-positive cancers [65]. The mechanism of action of this theranostic is as follows: vintafolide minimizes the off-target toxicity by delivering the vinca molecule directly and specifically to cancer cells that over-express the folate-receptor [66]. Once delivered to the cancer cell surface, Vintafolide is internalized into the cancer cell via endocytosis, a natural cellular process. Once inside the cell, Endocyte’s proprietary linker technology releases the chemotherapy to eliminate the cancer cell. In preclinical models, vintafolide demonstrated potent antitumour activity (IC_50_ 1–10 nM), exhibiting sparing effects on healthy tissues.

### 3.3. Nanotheranostics—Prerequisites for Developments

The impetus for developing antitumour drugs and treatment methods, which have become standard protocols still used today, was given in 1971 with the introduction of the National Cancer Act and the expansion of the National Cancer Institute’s (NCI) powers [67]. Many revolutionary discoveries in cancer molecular biology followed: restriction enzymes passage of national cancer act (1971); hybridomas and monoclonal antibodies 50% tracking of cancer statistics by seer program (1975); cellular origin of retroviral oncogenes (1979); epidermal growth factor and receptor 1981 suppression of tumor growth by p53 (1984); G proteins and cell (1984), etc. [68]. However, it took another 25 years of work before cancer treatment using nanoparticles as targeted delivery vehicles for diagnosis and therapy was realized: the first FDA-approved nanoparticle-based cancer drug was liposomal Doxil^®^ in 1995 (Figure 12).

The goal of developing liposomal Doxil^®^ was to reduce the side effects of doxorubicin therapy, such as cardiomyopathy, by targeted delivery to the organ of interest [69,70]. Components of the liposomal dosage form (see Figure 9) collectively facilitate physiological endocytosis, endosomal escape, and release of the API cargo into the cytosol for translation [71].

Extensive experiments were conducted to isolate, stabilize, and study the physicochemical properties of nanoparticles whose sizes are comparable to the de Broglie wavelength of their charge carriers (i.e., electrons and holes) [72]. When this condition is met, a quantum effect occurs, whereby particles behave like zero-dimensional quantum dots that obey the rules of quantum mechanics [73]. In nanocrystals, electron wave functions are confined due to the increasing discreteness (lack of continuity) of energy levels, unlike bulk material of the same substance, resulting in higher energy and wider band gaps (Equation (4)):(4)$$\Delta E=\;\frac{n^{2}h^{2}}{{8ma}^{2}}$$
where ΔE is the energy shift between discrete and continuous electrons in nano- and bulk-sized materials; *n*^2^ is the principal quantum number; *h* is Planck’s constant; *m* is the effective mass, and *a* is the quantum dot radius.

Special optical properties, caused by surface plasmon excitation in metallic nanoparticles and their ability to self-organize, can be exploited in medicine.

Key postulates and discoveries for future nanomedicine include the paradigm of assembling atoms into particles with the possibility of directed manipulation (“plenty of room at the bottom”), proposed by Richard Feynman in 1959 [74,75]; the discovery of liposomes by Alec Bangham in mid-1960s [76,77]; the introduction of the term “nanotechnology” by Norio Taniguchi in 1974 [78]. According to The National Nanotechnology Initiative (NNI), nanotechnology («nano» from the Greek word means «dwarf») is “a science, engineering, and technology conducted at the nanoscale (1 to 100 nm), where unique phenomena enable novel applications in a wide range of fields, from chemistry, physics and biology, to medicine, engineering and electronics” [79].

The dawn of the era of nanocytostatics (NCTCs) dictates the following characteristics for drugs to be effective in treatment: nanocarriers must reach the tumor with adequate API content; the drug pharmacokinetics (PK), biodistribution (BD), should be controlled by the nanocarriers and demonstrate a highly prolonged plasma circulation time; NCTCs should be available to tumor cells either by drug release at the tumor site or by the nanocarriers internalizing with the drug into tumor cells [80].

#### 3.3.1. Radioactive Nanotheranostics

Attachment of therapeutic radionuclides to liposomes has shown significant promise in cancer treatment [78]. Based on the needs of modern oncological medicine, which include developing effective methods of noninvasively tracking and quantifying the distribution of liposomes, as targeted delivery vehicles for APIs in the body, methods for labeling liposomes with theranostic radionuclides have been proposed [79]. Nuclear methods, including radionuclide techniques, encompass highly sensitive modalities such as positron emission tomography (PET), gamma-emitting techniques such as single-photon emission tomography (SPECT), and planar scintigraphy. Scintigraphy (oт лaт. “scintilla”—spark or flicker) involves visualizing target liposomes in vivo using externally placed nuclear cameras.

The most commonly used nuclide for radiolabeling liposomes is Technetium-99m (^99m^Tc). Its advantages include availability, relatively low cost, imaging capability, and an optimal half-life (see Table 2), and allowing imaging over 24 h [80].

Technetium-99m is a decay product of molybdenum-99 and undergoes gamma decay to form the ground state of technetium-99. Technetium-99 in the ground state can further decay to ruthenium (element 44). The overall synthesis and decay scheme is shown in (Equation (5)):(5)γ decay M4299o⟶T4399 mc⟶T4399c⟶R4499u67 h6 h211,000 years

The second most widely used radionuclide for liposome radiolabeling is Indium-111 (^111^In), followed by iodine radioisotopes (see Table 2).

Depending on the method of radionuclide attachment, surface and internal nanoscale liposomal targeted systems are distinguished (Figure 13).

#### 3.3.2. Nanoparticles

Radioactive nanoparticles have found a unique application as an inhalable nanoaerosol containing radionuclides. The advantage of using radioactive nanoparticles is the ability to increase the effectiveness of lung imaging, using computed tomography (CT), magnetic resonance imaging (MRI), positron emission tomography (PET), and single-photon emission computed tomography (SPECT) as a standard method in lung disease diagnosis [81]. Examples of nanoparticles used for this purpose in medicine are presented in Table 3 [82,83,84].

The examples of metallic nanoparticles of gold, silver, or iron (III) oxide presented in Table 3 have also attracted attention due to their potential for drug delivery (Au, Ag) and imaging (Fe) [85,86,87,88,89]. Metallic and oxide nanoparticles possess unique advantages described above, as well as a high surface area-to-volume ratio and the ability to penetrate biological barriers. Several formulations based on metallic nanoparticles have undergone clinical trials, and some have already been approved by the FDA, such as NanoTherm^®^ (superparamagnetic iron oxide nanoparticles—SPION https://www.nanothermtx.com/, accessed on 12 June 2025) for the treatment of glioblastoma using magnetic hyperthermia (induction) therapy. Equipment for magnetic induction hyperthermia (MIH) converts magnetic energy into heat. This approach is considered an effective “green” cancer therapy due to its high safety and efficacy, attributable to the greater sensitivity of cancer cells to temperature compared to normal cells: cancer cells can be destroyed when the temperature reaches the target range of 42–46 °C [90].

Gold nanoparticles (AuNPs) are regarded as active components in cancer therapy and photothermal treatment (thermal ablation, TrAbl): AuNPs absorb electromagnetic radiation (near-infrared light, λ = 650–950 nm), acting as a heat source [91,92]. The heat flux density from the nano-source depends on the local electromagnetic field (Equation (6)):(6)$$Q\;(x)\;=\;\frac{w}{2}e_{0}Im(e_{r}(x)){|E(x)|}^{2},\;\forall x\;\in \;\Omega$$
where *Ex* (electromagnetic field) is the solution of the Helmholtz equation with a radiation boundary condition; ω is the incident angular frequency; *e*_0_ is the permittivity of a vacuum; and *e_r_* is the complex relative permittivity of materials.

The efficiency of ablation is directly related to the size, shape, and agglomeration of nanoparticles (within the de Broglie wavelength) and their therapeutic concentration. However, as the size increases, so does the likelihood of damage to healthy tissues surrounding the tumor.

Regarding silver nanoparticles (AgNPs), according to research findings [93], AgNPs act as inducers of reactive oxygen species (ROS) within cells, causing oxidative stress through lipid peroxidation. Intracellular ROS production induced by AgNPs should be considered a key indicator of toxicity and may be viewed as the initial step in toxicity cascades: due to depletion of antioxidant capability (DAC), cells undergo programmed cell death (apoptosis) at the initiation stage of the toxicity cascade [94].

Radiotheranostics, which combines the diagnostic action of unstable nuclei with therapeutic effects, serves as an effective adjunct and/or standalone method in the treatment of oncology patients. Despite emerging resistance to both chemo- and radiotherapy, existing radionuclide resources provide protection against subsequent tumor metastasis. However, given the unfavorable prognosis for cancer incidence over the next 25 years, the development of “ahead-of-the-curve” pharmaceuticals is welcomed. This will be facilitated by current medical knowledge and understanding of ligand–receptor interaction mechanisms to trigger apoptosis in rapidly proliferating cells, as detailed in this research review.

## 4. Future Directions

The development of antitumour drugs for targeted delivery to organs expressing higher levels of endocytic receptors represents a complex set of challenges related to resistance, limitations, and opportunities that affect their efficacy. One promising avenue for cancer therapy, beyond the identification and study of new nuclear isotopes [95,96], is the development and application of genetically modified and unmodified oncolytic viruses capable of also targeting the tumor microenvironment [97,98].

All new methods introduced into cancer therapy promise to be as effective as possible in eradicating all cancer types [99]. However, experience shows that a universal «cure for cancer» is unlikely ever to be found [100]. Yet, in the hands of scientists, the development of adjunct therapies to existing treatments can gradually reduce oncogenicity and contribute to improved survival rates.

## Figures and Tables

**Figure 1 medsci-13-00107-f001:**
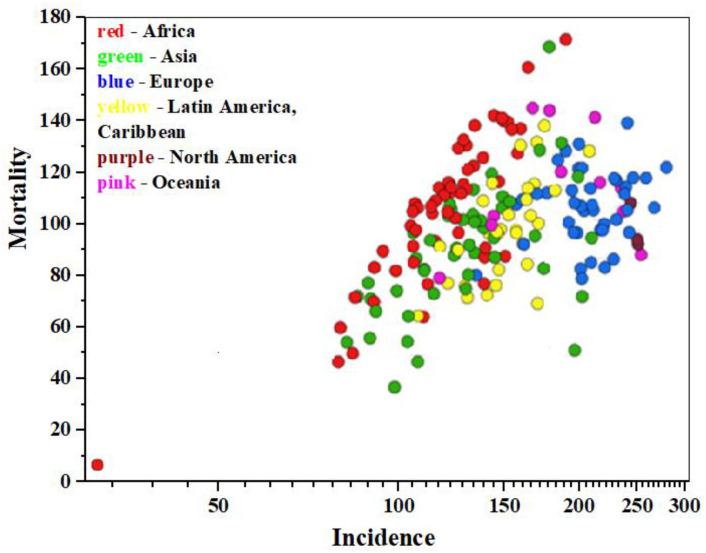
The epidemiology diagram of «Mortality—Incidence» (age-standardized mortality rate per 100.000 population), based on 2022 IARC data, for both sexes, all cancers except NMSCs.

**Figure 2 medsci-13-00107-f002:**
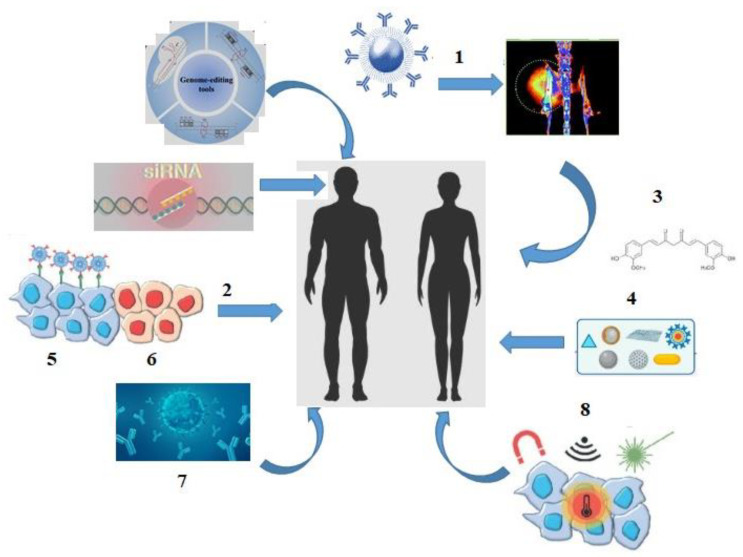
The «landscape» of cancer treatment in the 21st century: 1—theranostics; 2—targeted therapy; 3—antioxidants; 4—nanoparticles; 5—tumor cells; 6—healthy cells; 7—monoclonal antibodies (mABs); 8—thermal ablation, magnetic hyperthermia.

**Figure 3 medsci-13-00107-f003:**
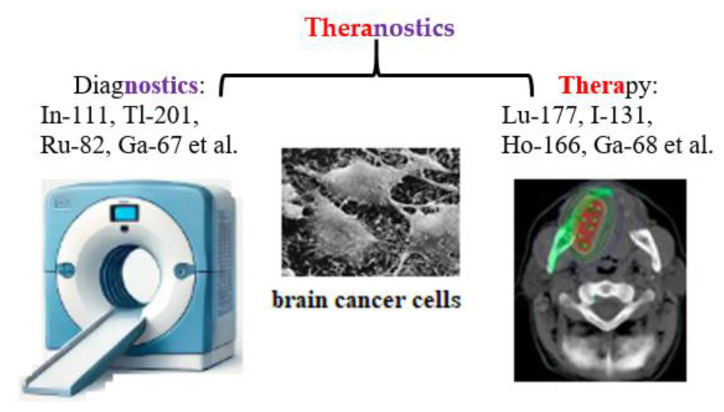
Schematic representation of theranostics, integrating diagnostics and therapeutics in nuclear medicine.

**Figure 4 medsci-13-00107-f004:**
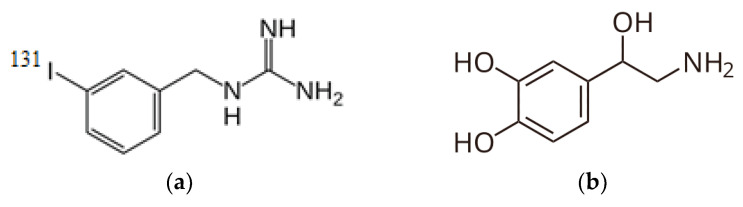
Radiopharmaceutical agent’s structural formulas: (**a**) MIBG and (**b**) Noradrenaline.

**Figure 5 medsci-13-00107-f005:**
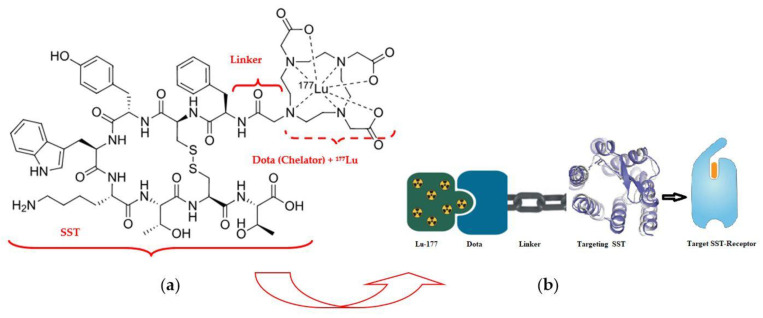
Radiopharmaceutical Lu-177 dota-tate agent: (**a**) structural formula; (**b**) demonstration of the lutetium-177 (^177^Lu)–Dotatate theranostic action.

**Figure 6 medsci-13-00107-f006:**
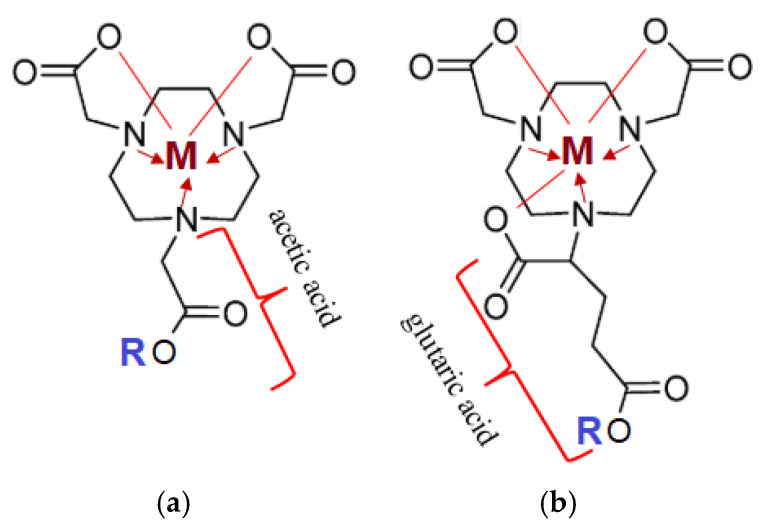
Structural representation of chelating agents with an ion of metal (M^+2,+3^: Lu-175, Ga-69, Ga-71): (**a**) NOTA (1,4,7-triazacyclononane-1,4,7-triacetic acid); (**b**) NODAGA (1,4,7-triazacyclononane,1-glutaric acid-4,7-acetic acid). The lines represent covalent polar bonds between the Me ion and the carboxyl group residue; the arrow represents covalent bonds via a donor-acceptor mechanism between a tertiary amine (donor) and the complexing agent (Me-acceptor).

**Figure 7 medsci-13-00107-f007:**
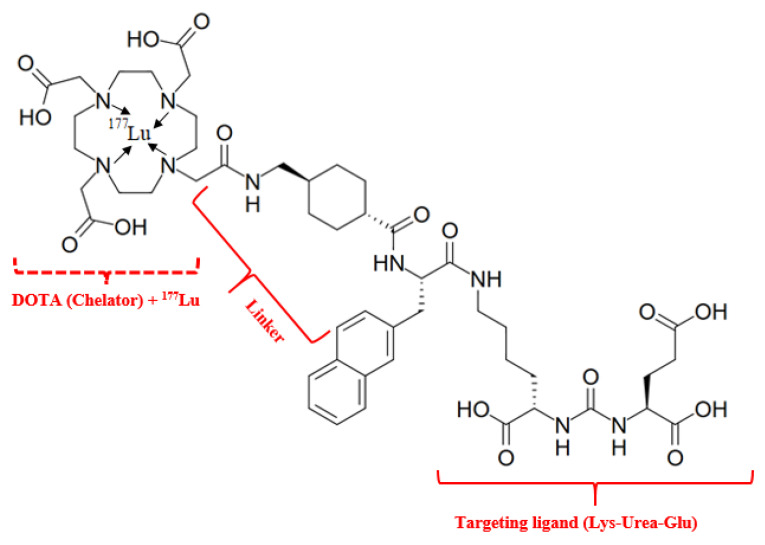
Radiopharmaceutical Lu-177 vipivotide tetraxetan. IUPAC Name: 2-[4-[2-[[4-[[(2S)-1-[[(5S)-5-carboxy-5-[[(1S)-1,3-dicarboxypropyl]carbamoylamino]pentyl]amino]-3-naphthalen-2-yl-1-oxopropan-2-yl]carbamoyl]cyclohexyl]methylamino]-2-oxoethyl]-7,10-bis(carboxylatomethyl)-1,4,7,10-tetrazacyclododec-1-yl]acetate;lutetium-177(3+).

**Figure 8 medsci-13-00107-f008:**
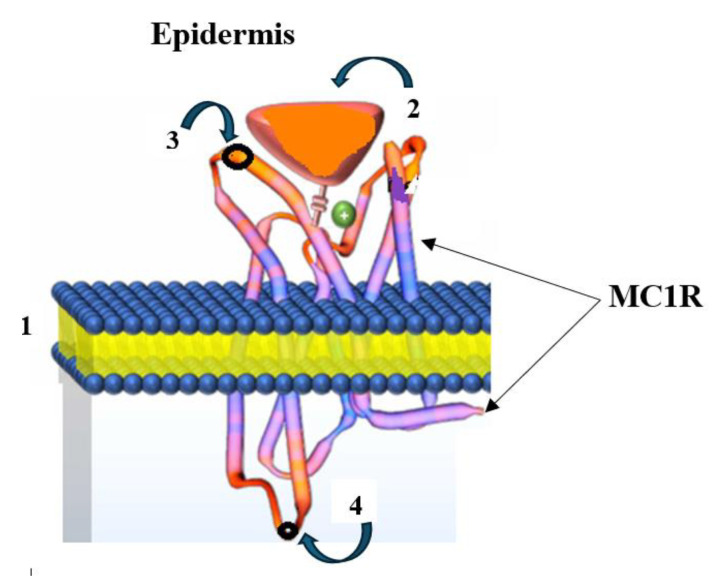
Structure of a melanocyte (from greek *μέλας*—“black” and *κύτος*—“cell”): 1—cell membrane; 2—α-melanocyte-stimulating hormone (α-MSH); 3—extracellular N-linked glycosylation site on the MC1R receptor’s extracellular terminus; 4—intracellular C-linked site.

**Figure 9 medsci-13-00107-f009:**
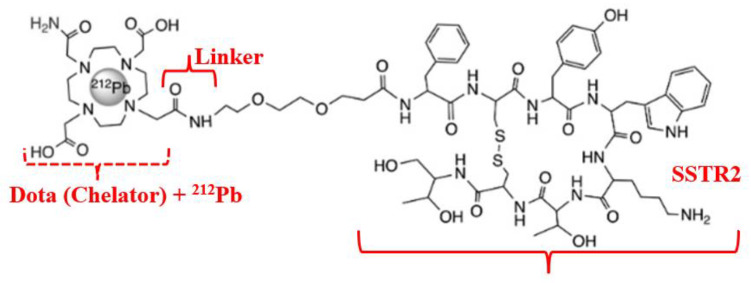
Radiopharmaceutical Pb-212 dota-tate agent for MC1R targeting.

**Figure 10 medsci-13-00107-f010:**
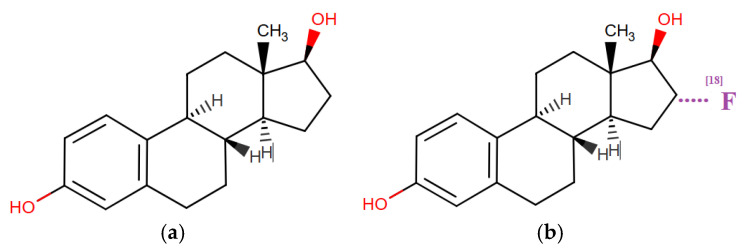
Molecular structures of APIs: (**a**) 17b-estradiol; (**b**) 6α-fluoro-17β-estradiol (^18^F-fluoroestradiol, FES). UPAC Name: (8R,9S,13S,14S,16R,17R)-16-(^18^F)fluoranyl-13-methyl-6,7,8,9,11,12,14,15,16,17-decahydrocyclopenta[a] phenanthrene-3,17-diol.

**Figure 11 medsci-13-00107-f011:**
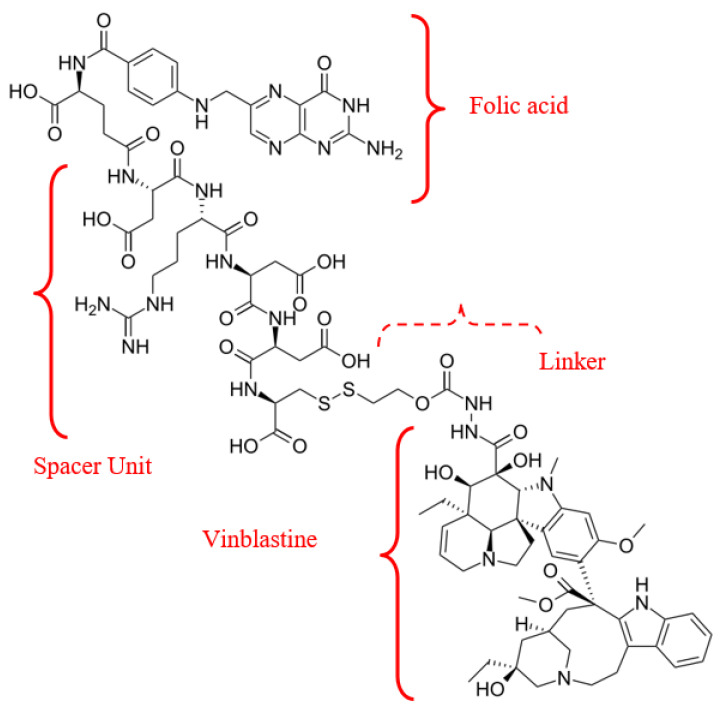
Radiopharmaceutical Vintafolide agent’s structural formula.

**Figure 12 medsci-13-00107-f012:**
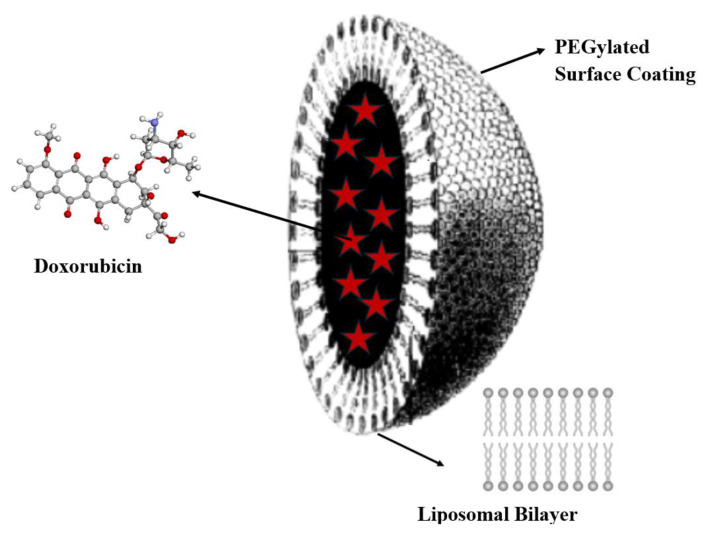
The structure of the Doxil^®^ liposomal dosage form, containing doxorubicin nanoencapsulated in a liquid vesicle core (red star symbols), stabilized with methoxypolyethylene glycol (d~80–100 nm).

**Figure 13 medsci-13-00107-f013:**
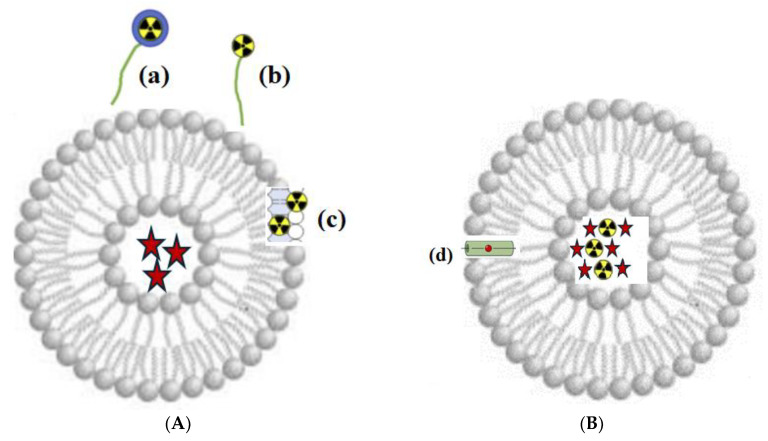
Principle of radioactive labeling of liposomes. (**A**) Surface radioactive labeling: (a) radionuclide with chelator (Me); (b) radionuclide without chelator, linked to the liposome membrane via a PEG chain; and (c) radionuclide embedded in the lipid bilayer. (**B**) Intraliposomal radioactive labeling: radionuclide (black and yellow) and API (red star symbols), encapsulated in the aqueous core; (d) ionophore channel for transporting radionuclides across the bilayer.

**Table 1 medsci-13-00107-t001:** The types of radiation for targeted therapy.

Radiation Type	Subatomic Particles	Mass × 10^−27^ kg	Energy	Tissue Penetration Range	Linear Energy Transfer, keV/µm	Relative Biological Effectiveness
Alpha, α *	2 protons and 2 neutrons	6.6	Discrete; ~4^–10^ MeV	~20–70 µm	~50–300	~5
Beta minus, β^−^ **	Electron	0.00091	Continuous; maximum of some hundred keV to some MeV	<1 mm	~0.1–2	~1
Auger ***	Electron	0.00091	<10 keV	<1 µm	~4–26	~1 or higher

* Alpha (α)—ionizing radiation consisting of positively charged particles formed by the nucleus of the atom He24+2 [24]. ** Beta minus, β^−^—ionizing radiation consisting of positively charged particles formed by the nucleus of the atom [25]. *** Auger electrons (e_A_) carry very low energy, emitted by unstable nuclei that decay by capturing electrons; the energy (e_A_) is transferred over distances from 1 × 10^−9^ to 1 × 10^−6^ m with low ionization energy loss per unit distance, causing the death of tumor cells [26].

**Table 2 medsci-13-00107-t002:** Radiopharmaceuticals.

Radionuclide				
Isotope Type	T_1/2_	Radioactive Decay (Electron Emission) *	Reactor (Therapy, Diagnosis)	Cyclotron (Diagnosis)
			Medical Application	
Mo-99	65.94 h	β^−^	diagnosing diseases—e.g., heart failure, cancer—with Tc-99m **	
Tc-99m	6 h	^99m^Tc → ^99^ Tc + γ (88%) IT (Isomeric Transition): ^99m^Tc → ^99^Tc^+^ + e^−^ (12%)	-	diagnoses of peptide, small molecule, and cell labeling, perfusion imaging
In-111	2.8047 days	e_A_	-	diagnoses of brain and colon
Xe-133	5.243 days	e_A_	lung ventilation studies	-
I-123	13.27 h	e_A_	-	diagnoses of thyroid function
Ho-166	26.8 h	β^−^	therapy of liver tumors	-
Tl-201	72.912 h	c_e_	-	detecting cardiac conditions
Lu-177	6.65 days	β^−^	therapy of neuroendocrine tumors	-
Ru-82	1.3 min	β^−^	-	detecting cardiac conditions
Ra-223	11.4 days	α	-	treatments for prostate cancer spread to bones
F-18	109.77 min	β^+^	-	visualization of tumors various localizations differential diagnostics
I-125 I-131	59.402 and 8.02070 days	e_A_ + c_e_ and c_e_	therapy of prostate cancer and thyroid conditions	-
Co-57	271.79 days	e_A_ + c_e_		
At-211	7.214 h	α	high degree of selectivity of therapeutic effect on thyroid tumor tissue (Isotope Separator On-Line Detector, ISOLDE) [32,33]	
Ga-67	3.2612 days	e_A_	-	diagnoses of infections and inflammation
Ga-68	67.71 min	β^+^	-	one of the first radiopharmaceutical markers (1963); visualization of prostate cancer, neuroendocrine tumors and other diseases
Cu-62	9.67 min	β^−^ + e_A_		
Cu-64	12.70 h	β^+^; β^−^ + e_A_		
Sr-89	50.53 days	β^−^	pain management in bone cancer	-
Y-90	64.10 h	β^−^	therapy of liver cancer and rheumatic conditions	-
Ir-192	73.827 days	β^−^	therapy of cervical, prostate, lung, breast and skin cancer	-
Pb-212	10.64 h	β^−^ and α	treatment of ovarian cancer and neuroendocrine tumors	-
Pb-203	51.873 h	e_A_	-	cancer marker

* β^−^—electron; e_A_—Auger electron; c_e_—conversion electron. ** Tc-99m—is a metastable nuclear isomer, as indicated by the “m” after its mass number 99.

**Table 3 medsci-13-00107-t003:** Examples of radioactive nanoparticles used in imaging and therapy, with their diagnostic modality and associated advantages/disadvantages.

Radioactive Nanoparticles	Radioactive Decay	Standard Method of Diagnosis	Advantages	Disadvantages
Tracing agents				
Tc-99m	γ (140 keV)	single-photon emission computed tomography	Trace amount is required	Radiations, limited spatial resolution (~15 mm), no lung morphology image
Ga-67	β^+^ (1899 keV)	positron emission tomography		Radiations, limited spatial resolution (~6 mm), no lung morphology image
AuNPs	X-ray absorption	comprising computed tomography	Therapeutic effect (photothermal and radiosensitiser)	higher concentration is required compared to other CAs
AgNPs			Antimicrobial activity	Easy to be deposited in other tissues after pulmonary delivery
Contrast agents				
FeNPs	Shortening the T1 relaxation time of nearby water	magnetic resonance imaging (MRI)	Magnetic hyperthermia therapy	Inflammatory response and extrapulmonary toxicity were observed upon inhalation
GdNPs	Shortening the T1 and T2 relaxation time of nearby water		Radiosensitiser	Toxicity of free ions
MnNPs	Shortening the T1 relaxation time of nearby water		Enhancement of photo- and chemotherapy	Neurotoxicity upon inhalation

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
