# Peer review of "The Novel Achievements in Oncological Metabolic Radio-Therapy: Isotope Technologies, Targeted Theranostics, Translational Oncology Research"

_medsci, 2025, doi:10.3390/medsci13030107_

Round 1

Reviewer 1 Report

Comments and Suggestions for Authors

I had to review the article :The Novel Achievements in Cancer Radiotherapy: Isotope Techologies, Targeted Theranostics, Translational Oncology Research

The article presents a review of medical science knowledge  in regard to novel cancer achievements in radiotherapy, commonly used radionuclides in radiotherapy for both treatment and diagnosis.

The article is well-structured and includes the main types of radiation for targeted therapy, radiopharmaceuticals- antimetabolites and  metabolites. Radioactive nanoparticles used in imaging and therapy are described in detail  with their diagnostic modality and associated advantages and disadvantages.

The bibliography is current and includes 100 titles and the future research directions are also outlined and mapped out.

I congratulate the authors for their comprehensive and up-to-date work.

Author Response

Response to Reviewer 1 Comments

The authors thank the respected Reviewer for valuable comments aimed at eliminating erroneous conclusions, inaccuracies in wording and at improving the quality of the material of the article «The Novel Achievements in Cancer Radiotherapy: Isotope Technologies, Targeted Theranostics, Translational Oncology Research» in accordance with the high requirements of Medical Sciences.

Comments : The article presents a review of medical science knowledge in regard to novel cancer achievements in radiotherapy, commonly used radionuclides in radiotherapy for both treatment and diagnosis:

  1. The article is well-structured and includes the main types of radiation for targeted therapy, radiopharmaceuticals- antimetabolites and metabolites. Radioactive nanoparticles used in imaging and therapy are described in detail with their diagnostic modality and associated advantages and disadvantages.
  2. The bibliography is current and includes 100 titles and the future research directions are also outlined and mapped out.
  3. I congratulate the authors for their comprehensive and up-to-date work.

Response 1: We express our sincere gratitude and deep appreciation to the highly respected Reviewer for this assessment of the quality and significance of our manuscript, which we hope will help oncologists and radiotherapists in their invaluable work to save seriously ill patients.

With gratitude, the authors

Reviewer 2 Report

Comments and Suggestions for Authors

First of all, I consider this review to be an ambitious attempt to summarize current developments in theranostics.
However, there are both structural and informational shortcomings:
- the title - The Novel Achievements in Cancer Radiotherapy - is vague and misleading. You talk only about metabolic radiotherapy. And "Cancer Radiotherapy" is not quite an academic term. I would recommend changing "The Novel Achievements in Cancer Radiotherapy" to "Novel Achievements in Oncological Metabolic Radiotherapy"
- the abstract presents a "Methods" section that is not satisfactory. The 'in extenso' manuscript has no "Methods" section. Please describe how you gathered the cited articles and what was the algorithm you used to write this paper. You even jump from 1. Introduction to 3. Results, suggesting that you forgot to add section 2.
- the abstract does not correctly summarise the content of the manuscript. I strongly recommend you rewrite the abstract in accordance with the manuscript. 
- please consider adding data about 177 Lu–PSMA (Lutetium-177–Prostate-Specific Membrane Antigen) therapy (Pluvicto). As a form of targeted radioligand therapy, 177 Lu–PSMA has recently gained clinical approval for the treatment of patients with metastatic castration-resistant prostate cancer (mCRPC). Given its growing relevance in the field of theranostics, the inclusion of Lu-PSMA would add more value.
- given the title ("Translational Oncology Research"), I think you should talk more about the clinical implications of metabolic radiotherapy in conjunction with other therapies. A good example of structure and therapies can be found in this article https://www.mdpi.com/2077-0383/14/3/912 - which can be used for a better example of how your reserach can be integrated in real-world clinician activity. 

This manuscript has a good potential of discussing emerging directions in metabolic radiotherapy, isotope technologies, and theranostic approaches in oncology. The effort to integrate historical background, physical principles, clinical applications, and translational research is valuable, but it needs more work.  

Author Response

Response to Reviewer 2 Comments

The authors thank the respected Reviewer for valuable comments aimed at eliminating erroneous conclusions, inaccuracies in wording and at improving the quality of the material of the article «The Novel Achievements in Cancer Radiotherapy: Isotope Technologies, Targeted Theranostics, Translational Oncology Research» in accordance with the high requirements of Medical Sciences.

Comments 1: First of all, I consider this review to be an ambitious attempt to summarize current developments in theranostics.

Response 1: We express our heartfelt gratitude to the highly respected Reviewer for his opinion about the authors' desire to achieve significant goals in working on the Manuscript. We were encouraged by the hope that the contents of this Review Article may be useful to treating oncologists and radiation therapists in saving seriously ill patients.

Comments 2: The title «The Novel Achievements in Cancer Radiotherapy…» - is vague and misleading. You talk only about metabolic radiotherapy. And "Cancer Radiotherapy" is not quite an academic term. I would recommend changing "The Novel Achievements in Cancer Radiotherapy" to "Novel Achievements in Oncological Metabolic Radiotherapy".

Response 2: Thank you very much! We fully agree with the opinion of the specialist, in connection with which the title of the manuscript has been changed to the following: «The Novel Achievements in Oncological Metabolic Radiotherapy: Isotope Technologies, Targeted Theranostics, Translational Oncology Research»

Comments 3: The abstract presents a "Methods" section that is not satisfactory. The 'in extenso' manuscript has no "Methods" section. Please describe how you gathered the cited articles and what was the algorithm you used to write this paper. You even jump from 1. Introduction to 3. Results, suggesting that you forgot to add section 2.

Response 3: Thank you for your fair comment. Corrected:

  1. Methods (Page 4).

The methodology for this review article involved developing our own search algorithm, which was based on the selection and analysis of information from the PubMed, Scopus, and Web of Science databases over the past 20 years. This approach was guided by the concept of the existing "structure-mechanism of action-pharmacological effect" relationship of active pharmaceutical ingredients, encompassing various forms such as atoms, solid-phase (nano-) objects (d = 1-100 nm), molecules, and complex particles. In addition to review articles, practical research, and clinical cases, data from accessible official online sources were also considered, including: the International Agency for Research on Cancer, the GLOBOCAN project, the National Human Genome Research Institute, the International Atomic Energy Agency, the Nuclear Research and Consultancy Group, Melanoma Unit, and NanoTherm.

Among the literary sources reviewed, articles from the last 5 years constituted approximately 60%, those from the last 20 years accounted for about 20%, and approximately 20% were older sources deemed valuable for the history of medicine.

The selection process involved screening abstracts and, more extensively, the full texts of articles. The final presentation of the material was conducted using a narrative synthesis approach, which included a systematic review and generalization of results from multiple studies, accompanied by our own conclusions. This approach was also applied to the presentation of graphic material.

Methods (Abstract Page 1).

The methodology involved searching, selecting, and analyzing information from PubMed, Scopus, and Web of Science databases, as well as from available official online sources over the past 20 years. The search was structured around the structure-mechanism-effect relationship of active pharmaceutical ingredients (APIs). The manuscript, including graphic materials, was prepared using a narrative synthesis method.

Comments 4: The abstract does not correctly summarise the content of the manuscript. I strongly recommend you rewrite the abstract in accordance with the manuscript.

Response 4: Thank you very much! Fixed in Abstract:

Background/Objectives. This manuscript presents an overview of advances in oncological radiotherapy as an effective treatment method for malignant tumors, focusing on mechanisms of action within metabolite-antimetabolite systems. The urgency of this topic is underscored by the fact that cancer remains one of the leading causes of death worldwide: as of 2022, approximately 20 million new cases were diagnosed globally, accounting for about 0.25% of the total population. Given prognostic models predicting a steady increase in cancer incidence to 35 million cases by 2050, there is an urgent need for the latest developments in physics, chemistry, molecular biology, pharmacy, and strict adherence to oncological vigilance. The purpose of this work is to demonstrate the relationship between the nature and mechanisms of past diagnostic and therapeutic oncology approaches, their current improvements, and future prospects. Particular emphasis is placed on isotope technologies in the production of therapeutic nuclides, focusing on the mechanisms of formation of simple and complex theranostic compounds and their classification according to target specificity. Methods. The methodology involved searching, selecting, and analyzing information from PubMed, Scopus, and Web of Science databases, as well as from available official online sources over the past 20 years. The search was structured around the structure-mechanism-effect relationship of active pharmaceutical ingredients (APIs). The manuscript, including graphic materials, was prepared using a narrative synthesis method. Results. The results present a sequential analysis of materials related to isotope technology, particularly nucleus stability and instability. An explanation of theranostic principles enabled a detailed description of the action mechanisms of radiopharmaceuticals on various receptors within the metabolite-antimetabolite system using specific drug models. Attention is also given to radioactive nanotheranostics, exemplified by the mechanisms of action of radioactive nanoparticles such as Tc-99m, AuNPs, AgNPs, FeNPs, and others. Conclusions. Radiotheranostics, which combines the diagnostic properties of unstable nuclei with therapeutic effects, serves as an effective adjunctive and/or independent method for treating cancer patients. Despite the emergence of resistance to both chemotherapy and radiotherapy, existing nuclide resources provide protection against subsequent tumor metastasis. However, given the unfavorable cancer incidence prognosis over the next 25 years, the development of "preventive" drugs is recommended. Progress in this area will be facilitated by modern medical knowledge and a deeper understanding of ligand-receptor interactions to trigger apoptosis in rapidly proliferating cells.

Comments 5: Please consider adding data about 177 Lu–PSMA (Lutetium-177–Prostate-Specific Membrane Antigen) therapy (Pluvicto). As a form of targeted radioligand therapy, 177 Lu–PSMA has recently gained clinical approval for the treatment of patients with metastatic castration-resistant prostate cancer (mCRPC). Given its growing relevance in the field of theranostics, the inclusion of Lu-PSMA would add more value.

Response 5: Thank you for your fair comment. Corrected (Pages 9-10).):

177Lu—vipivotide tetraxetan RLT

Lutetium-177 vipivotide tetraxetan (PluvictoR) ("vipivo"- targeting moiety Lys-Urea-Glu, the "tide" suffix - peptide nature of this moiety and "tetraxetan" is a DOTA derived from) - is a RLT drug first approved by the FDA on March 23, 2022, for the treatment of prostate-specific membrane antigen-positive metastatic castration-resistant prostate cancer (NIH, https://www.cancer.gov/publications/dictionaries/cancer-drug/def/lutetium-lu-177-vipivotide-tetraxetan). It is generally accepted that the mechanism of action of 177Lu-PSMA-617 is attributed to its radioligand activity [50.          Keam, S.J. Lutetium Lu 177 Vipivotide Tetraxetan: First Approval. Mol Diagn Ther 2022, 26(4), 467-475]. The structure comprises the main fragments: radionuclide (Lu-177) - a source of β radiation; targeting ligand (vipivotide) - a PSMA-binding peptide (Lys-Urea-Glu) that specifically targets prostate cancer cells, as PSMA is significantly overexpressed on their surface; chelator (tetraxetan) - a chemical moiety that securely binds the radionuclide to the targeting ligand; hydrophobic linker, composed of 2-naphthyl-L-Ala and cyclohexyl groups, it connects the targeting ligand to the chelator, influencing the compound's pharmacological properties (Figure 7).

Figure 7. Radiopharmaceutical Lu-177 vipivotide tetraxetan*.

*IUPAC Name: 2-[4-[2-[[4-[[(2S)-1-[[(5S)-5-carboxy-5-[[(1S)-1,3-dicarboxypropyl]carbamoylamino]pentyl]amino]-3-naphthalen-2-yl-1-oxopropan-2-yl]carbamoyl]cyclohexyl]methylamino]-2-oxoethyl]-7,10-bis(carboxylatomethyl)-1,4,7,10-tetrazacyclododec-1-yl]acetate;lutetium-177(3+)

Comments 6: Given the title ("Translational Oncology Research"), I think you should talk more about the clinical implications of metabolic radiotherapy in conjunction with other therapies. A good example of structure and therapies can be found in this article https://www.mdpi.com/2077-0383/14/3/912 - which can be used for a better example of how your reserach can be integrated in real-world clinician activity.

Response 6: Thank you for your suggestion for an addition. It was done (Page 8).

«Combination of metabolic radiotherapy (MtRth) with other treatment methods is the most important direction for increasing the efficiency of cancer treatment and reducing the frequency of side effects in clinical practice. This goal can be achieved by combining MtRth with neoadjuvant (NCRT) and adjuvant chemoradiotherapy (ACRT) before surgery, for example, immunotherapy, proton beam therapy, hyperthermia, etc. [41. LiÈ™cu, H.-D.; Verga, N.; Atasiei, D.-I.; Ilie, A.-T.; Vrabie, M.; RoÈ™u, L.; PoÈ™taru, A.; Glăvan, S.; LucaÈ™, A.; Dinulescu, M.; et al. Therapeutic Management of Locally Advanced Rectal Cancer: Existing and Prospective Approaches. J. Clin. Med. 2025, 14, 912. https://doi.org/10.3390/jcm14030912].

 With gratitude, the authors

Reviewer 3 Report

Comments and Suggestions for Authors

Hello.
I believe that it is a well-conceived and supported work with well-argued evidence and supplemented with valuable bibliographic indexes.
Browsing the literature and comparing it with the data presented in the article, I believe that the information provided in the hope of publication is useful for oncologists and radiotherapists.
However, I would like to supplement the types of pathologies presented, and I suggest that more clinical applicability of the data presented be remembered and described, with the favorable results obtained following the treatment, but also the complications that the application of these treatments implies. Perhaps a short discussion chapter would be useful with the presentation of these data
I congratulate the authors for the complexity of the work and the data provided.

Author Response

Response to Reviewer 3 Comments

The authors thank the respected Reviewer for valuable comments aimed at eliminating erroneous conclusions, inaccuracies in wording and at improving the quality of the material of the article «The Novel Achievements in Cancer Radiotherapy: Isotope Technologies, Targeted Theranostics, Translational Oncology Research» in accordance with the high requirements of Medical Sciences.

Comments 1: I believe that it is a well-conceived and supported work with well-argued evidence and supplemented with valuable bibliographic indexes. Browsing the literature and comparing it with the data presented in the article, I believe that the information provided in the hope of publication is useful for oncologists and radiotherapists.

Response 1: The authors sincerely thank the highly respected reviewer for the impression the contents of our manuscript made on him!

Comments 2: However, I would like to supplement the types of pathologies presented, and I suggest that more clinical applicability of the data presented be remembered and described, with the favorable results obtained following the treatment, but also the complications that the application of these treatments implies. Perhaps a short discussion chapter would be useful with the presentation of these data

Response 2: Thank you very much! We fully agree with the opinion of the specialist, in connection with which the some chapters of the manuscript has been changed to the following:

The 3.2.2. Chapter (Page 9):

Overall, PRRT with 177Lu-Dotatate is an effective treatment for patients with progressive, high-grade NETs, with an approximately 80% reduction in the risk of progression or death. Monitoring of renal and blood function during and after therapy is recommended to minimize this risk [49.Strosberg, J.; El-Haddad, G.; Wolin, E.; Hendifar, A.; Yao, J.; et. al. NETTER-1 Trial Investigators. Phase 3 Trial of 177Lu-Dotatate for Midgut Neuroendocrine Tumors. N Engl J Med 2017, 12, 376(2):125-135].

The «177Lu—vipivotide tetraxetan RLT» Chapter (Page 10):

The results of 177Lu-vipivotide tetraxetan therapy show a pronounced biochemical (decrease in the level of total PSA) response, as well as low toxicity (often in the form of the development of grade I xerostomia) [51. Li A.A., Geliashvili T.M., Rumyantsev A.A. et al. Impressive response to 177Lu-PSMA-617 therapy in a patient with metastatic castration-resistant prostate cancer refractory to apalutamide, docetaxel and metastasis-directed therapy. Onkourologiya = Cancer Urology 2024;20(4):98–103. (In Russ.)].

The 3.2.3. Radionuclide therapy with hormone-receptors. 18F-Fluoroestradiol Chapter (Page 12):

18F-Fluoroestradiol has been used as a research agent since the 1980s and as a clinical agent since 2016 in France and 2020 in the United States, without any major adverse events reported to date [62. O'Brien SR, Edmonds CE, Lanzo SM, Weeks JK, Mankoff DA, Pantel AR. 18F-Fluoroestradiol: Current Applications and Future Directions. Radiographics. 2023 Mar;43(3):e220143].

Comments 3: I congratulate the authors for the complexity of the work and the data provided.

Response 3: The authors sincerely thank the highly respected reviewer for the high assessment of our work

 With gratitude, the authors

Round 2

Reviewer 2 Report

Comments and Suggestions for Authors

Most of my concerns were addressed accordingly. Well done!
I have one more issue with your text: you have many paragraphs which you cite with a website (such as rows 104, 133, 165, 280 etc). Please eliminate all links unless absolutely necessary for the text to be understood - and cite the websites / authors / publications accordingly. 

Comments on the Quality of English Language

Some phrases can be simplified.